# TLDR: Unsupervised Goal-Conditioned RL via Temporal Distance-Aware Representations

**Junik Bae**   **Kwanyoung Park**   **Youngwoon Lee**
Yonsei University
https://heatz123.github.io/tldr

**Abstract:** Unsupervised goal-conditioned reinforcement learning (GCRL) is a promising paradigm for developing diverse robotic skills without external supervision. However, existing unsupervised GCRL methods often struggle to cover a wide range of states in complex environments due to their limited exploration and sparse or noisy rewards for GCRL. To overcome these challenges, we propose a novel unsupervised GCRL method that leverages TemporaL Distance-aware Representations (TLDR). Based on temporal distance, TLDR selects faraway goals to initiate exploration and computes intrinsic exploration rewards and goal-reaching rewards. Specifically, our exploration policy seeks states with large temporal distances (i.e. covering a large state space), while the goal-conditioned policy learns to minimize the temporal distance to the goal (i.e. reaching the goal). Our results in six simulated locomotion environments demonstrate that TLDR significantly outperforms prior unsupervised GCRL methods in achieving a wide range of states.

**Keywords:** Unsupervised Goal-Conditioned Reinforcement Learning, Temporal Distance-Aware Representations

## 1 Introduction

Human babies can autonomously learn goal-reaching skills, starting from controlling their own bodies and gradually improving their capabilities to achieve more challenging goals, involving *longer-horizon* behaviors. Similarly, for intelligent agents like robots, the ability to reach a large set of states–including both the environment states and agent states–is crucial. This capability not only serves as a foundational skill set by itself but also enables achieving more complex tasks.

Can robots autonomously learn such long-horizon goal-reaching skills like humans? This is particularly compelling as learning goal-reaching behaviors in robots is task-agnostic and does not require any external supervision, offering a scalable approach for unsupervised pre-training of robots [3, 4, 5, 6, 7, 8, 9]. However, prior unsupervised goal-conditioned reinforcement learning (GCRL) [10, 2] and unsupervised skill discovery [1] methods exhibit limited coverage of reachable states in complex environments, as shown in Figure 1.

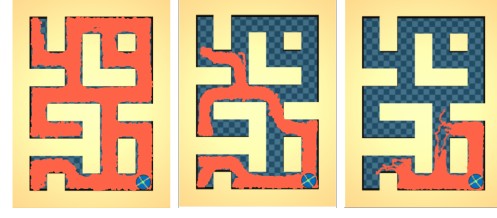

(a) TLDR (ours)   (b) METRA   (c) PEG

Figure 1: Trajectories (red) of an ant robot in a complex maze trained by TLDR, METRA [1], and PEG [2]. While prior methods yield limited exploration, **TLDR explores the entire maze**.

The major challenges in unsupervised GCRL are twofold: (1) exploring *diverse* states that the agent can learn to achieve, and (2) effectively learning a *goal-reaching policy*. Prior unsupervised GCRL methods focus on exploring novel states [11] or states with high uncertainty in next state prediction [10, 2]. However, discovering unseen states or state transitions may not lead to meaningful states.

8th Conference on Robot Learning (CoRL 2024), Munich, Germany.

Additionally, training a goal-reaching policy to maximize sparse [8] or heuristic [10, 12] goal-reaching rewards is often insufficient for long-horizon goal-reaching behaviors in complex environments.

In this paper, we propose a novel unsupervised GCRL method that leverages **TemporaL Distance-aware Representations (TLDR)** to improve both goal-directed exploration and goal-conditioned policy learning. TLDR uses temporal distance (i.e. the minimum number of environment steps between two states) induced by temporal distance-aware representations [1, 13, 14] for (1) selecting faraway goals to initiate exploration, (2) learning an exploration policy that maximizes temporal distance, and (3) learning a goal-conditioned policy that minimizes temporal distance to a goal.

TLDR demonstrates superior state coverage compared to prior unsupervised GCRL and skill discovery methods in complex AntMaze environments, as shown in Figure 1. Our ablation studies confirm that our temporal distance-aware approach enhances both goal-directed exploration and goal-conditioned policy learning. Furthermore, our method outperforms prior work across diverse locomotion environments, underscoring its general applicability.

## 2 Related Work

**Unsupervised goal-conditioned reinforcement learning (GCRL)** aims to learn a goal-conditioned policy that can reach diverse goal states without external supervision [15, 16, 10, 2]. The major challenges of unsupervised GCRL can be summarized in two aspects: (1) optimizing a goal-conditioned policy and (2) collecting trajectories with novel goals that effectively enlarge its state coverage.

To improve the efficiency of **goal-conditioned policy learning**, hindsight experience reply (HER) [8] and model-based policy optimization [10, 12] have been widely used. However, learning complex, long-horizon goal-reaching behaviors remains difficult due to sparse (e.g. whether it reaches the goal [8]) or heuristic rewards (e.g. cosine similarity between the state and goal [10, 12]).

Instead, temporal distance, defined as the number of environment steps between states estimated from data, can provide more dense and grounded rewards [17, 10, 18, 19]. LEXA [10] and PEG [2] use the expected temporal distances regarding the current policy as goal-reaching rewards [19]. However, this does not reflect the *"shortest temporal distance"* between states, often leading to sub-optimal goal-reaching behaviors. In this paper, we propose to use the estimated shortest temporal distance as reward signals for GCRL, inspired by QRL [14] and HILP [13]. We apply the learned representations to compute goal-reaching rewards rather than directly learning the value function in QRL or using it for skill-learning rewards in HILP.

**Exploration** in unsupervised GCRL relies heavily on selecting exploratory goals that lead an agent to novel states and expand state coverage. Exploratory goals can be simply sampled from a replay buffer as in LEXA [10], or can be selected from less visited states [20], states with low-density in state distributions [21, 11], and states with high uncertainty in dynamics [2]. Instead of sampling uncertain or less visited states as goals, we select states temporally distant from the visited state distribution as goals, encouraging the discovery of temporally farther away states.

In addition to exploratory goal selection, an explicit exploration policy [22, 20] can further encourage exploration by maximizing intrinsic rewards, such as uncertainty in dynamics used in LEXA and PEG. For better exploration, our approach opts for maximizing temporal distance from the visited states, continuously seeking novel and faraway states.

**Unsupervised skill discovery** [23, 24, 25, 26, 27, 28, 29, 1] is another approach to learning diverse behaviors without supervision, yet often lacks robust exploration capabilities [29], requiring manual feature engineering or limiting to low-dimensional state spaces. METRA [1] addresses these limitations by computing skill-learning rewards with temporal distance-aware representations. While achieving remarkable exploration and zero-shot goal-reaching capabilities, METRA exhibits limited coverage in complex environments, as depicted in Figure 1. We find that METRA tends to focus on reaching the known farthest states rather than exploring less visited states, whereas our exploration strategy explicitly encourages reaching unseen farther states.

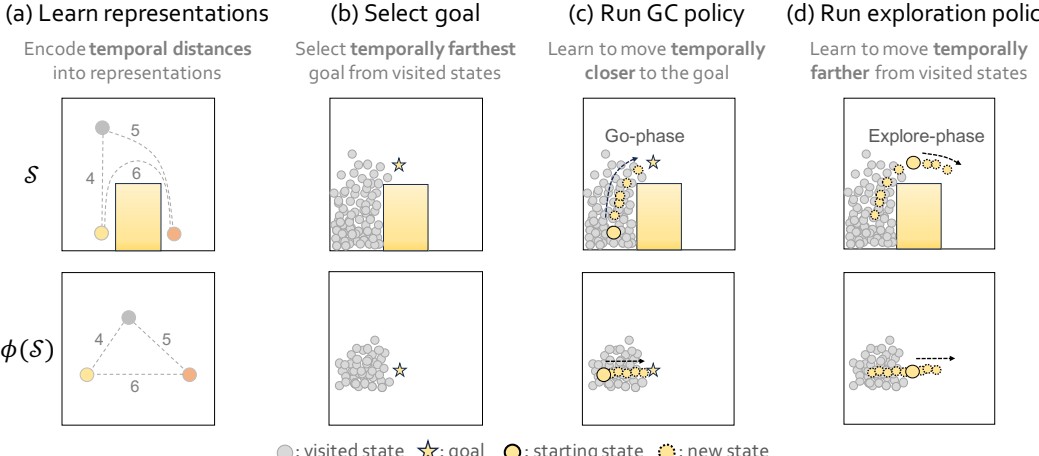

| (a) Learn representations | (b) Select goal | (c) Run GC policy | (d) Run exploration policy |

: visited state  ☆: goal  ◯: starting state  ◌: new state

Figure 2: **Overview of TLDR algorithm.** TLDR leverages temporal distance-aware representations for unsupervised GCRL. (a) We start by learning a state encoder $\phi(\mathbf{s})$ that maps states to temporal distance-aware representations. With the temporal distance-aware representations, TLDR (b) selects the *temporally* farthest state from the visited states as an exploratory goal, (c) reaches the chosen goal using a goal-conditioned policy, which learns to minimize temporal distance to the goal, and (d) collects exploratory trajectories using an exploration policy that visits states with large temporal distance from the visited states.

## 3   Approach

In this paper, we introduce **TemporaL Distance-aware Representations (TLDR)**, an unsupervised goal-conditioned reinforcement learning (GCRL) method, integrating *temporal distance*-aware representations (Section 3.2) into every facet of the Go-Explore strategy [20] (Section 3.3), as illustrated in Figure 2. TLDR first chooses a goal from experience (Section 3.4), reaches the selected goal via the goal-conditioned policy, and executes the exploration policy to gather diverse experiences. Both the exploration policy (Section 3.5) and goal-conditioned policy (Section 3.6) are then trained on the collected data and rewards computed using the temporal distance-aware representations. We describe the full algorithm in Algorithm 1 and implementation details in Appendix A.

### 3.1   Problem Formulation

We formulate the unsupervised GCRL problem with a goal-conditioned Markov decision process, defined as the tuple $\mathcal{M} = (\mathcal{S}, \mathcal{A}, p, \mathcal{G})$. $\mathcal{S}$ and $\mathcal{A}$ denote the state and action spaces, respectively. $p : \mathcal{S} \times \mathcal{A} \to \Delta(\mathcal{S})$ denotes the transition dynamics, where $\Delta(\mathcal{X})$ denotes the set of probability distributions over $\mathcal{X}$. The goal of the agent is to learn an optimal goal-conditioned policy $\pi^G : \mathcal{S} \times \mathcal{G} \to \mathcal{A}$, where $\pi^G(\mathbf{a} \mid \mathbf{s}, \mathbf{g})$ outputs an action $\mathbf{a} \in \mathcal{A}$ that can navigate to the goal $\mathbf{g} \in \mathcal{G}$ from the state $\mathbf{s}$ within minimum steps. In this paper, we set $\mathcal{G} = \mathcal{S}$, allowing any state as a potential goal for the agent.

### 3.2   Learning Temporal Distance-Aware Representations

Temporal distance, defined as the minimum number of environment steps between states, can provide more dense and grounded rewards for goal-conditioned policy learning as well as exploration. For GCRL, instead of relying on sparse and binary goal-reaching rewards, the change in temporal distance before and after taking an action can be an informative learning signal. Moreover, exploration in unsupervised GCRL can be incentivized by discovering temporally faraway states.

Therefore, in this paper, we propose to use temporal distance for unsupervised GCRL. We first estimate the temporal distance by learning temporal distance-aware representations, inspired by Park et al. [13], Wang et al. [14]. The learned representation $\phi : \mathcal{S} \to \mathcal{Z}$ encodes the temporal distance

**Algorithm 1** TLDR: unsupervised goal-conditioned reinforcement learning algorithm
───────────────────────────────────────────────────────────────────────
1: Initialize goal-conditioned policy $\pi_\theta^G$, exploration policy $\pi_\theta^E$, temporal distance-aware representation $\phi$, and replay buffer $\mathcal{D}$
2: **while** not converged **do**
3:     $\mathbf{s}_0 \sim p(\mathbf{s}_0)$
4:     Sample a minibatch $\mathcal{B} \sim \mathcal{D}$
5:     $\mathbf{g} \leftarrow \arg\max_{\mathbf{s}\in\mathcal{B}}(r_{\text{TLDR}}(\mathbf{s}))$             ▷ Select state with the highest TLDR reward (Eq. (2))
6:     **for** $t = 0, \ldots, T-1$ **do**
7:       **if** $t < T_G$ **then**
8:         $\mathbf{a}_t \sim \pi_\theta^G(\cdot \mid \mathbf{s}_t, \mathbf{g})$             ▷ Follow goal-conditioned policy $\pi_\theta^G$ for $T_G$ steps
9:       **else**
10:         $\mathbf{a}_t \sim \pi_\theta^E(\cdot \mid \mathbf{s}_t)$             ▷ Explore using exploration policy $\pi_\theta^E$
11:       $\mathbf{s}_{t+1} \sim p(\cdot \mid \mathbf{s}_t, \mathbf{a}_t)$
12:       $\mathcal{D} \leftarrow \mathcal{D} \cup \{\mathbf{s}_t, \mathbf{a}_t, \mathbf{s}_{t+1}\}$
13:     Train representations $\phi$ to minimize $\mathcal{L}_\phi$ in Eq. (1)
14:     Train exploration policy $\pi_\theta^E$ to maximize Eq. (3)
15:     Train goal-conditioned policy $\pi_\theta^G$ using HER with dense reward in Eq. (4)
───────────────────────────────────────────────────────────────────────

between two states into the latent space $\mathcal{Z}$, where $\|\phi(\mathbf{s}_1) - \phi(\mathbf{s}_2)\|$ represents the temporal distance between $\mathbf{s}_1$ and $\mathbf{s}_2$. This representation is then used across the entire unsupervised GCRL algorithm: exploratory goal selection, intrinsic reward for exploration, and reward for a goal-conditioned policy.

To train temporal distance-aware representations, we adopt QRL's constrained optimization [14]:

$$\max_\phi \mathbb{E}_{\mathbf{s}\sim p_{\mathbf{s}}, \mathbf{g}\sim p_{\mathbf{g}}} \left[ f(\|\phi(\mathbf{s}) - \phi(\mathbf{g})\|) \right] \quad \text{s.t.} \quad \mathbb{E}_{(\mathbf{s},\mathbf{a},\mathbf{s}')\sim p_{\text{transition}}} [\|\phi(\mathbf{s}) - \phi(\mathbf{s}')\|] \leq 1, \quad (1)$$

where $f$ is an affine-transformed softplus function that assigns lower weights to larger distances $\|\phi(\mathbf{s}) - \phi(\mathbf{g})\|$. We optimize this constrained objective using dual gradient descent with a Lagrange multiplier $\lambda$, and we randomly sample $\mathbf{s}$ and $\mathbf{g}$ from a minibatch during training.

### 3.3 Unsupervised GCRL with Temporal Distance-Aware Representations

With temporal distance-aware representations, we can integrate the concept of temporal distance into unsupervised GCRL. Our approach is built upon the Go-Explore procedure [20], a widely-used unsupervised GCRL algorithm comprising two phases: (1) the "**Go-phase**," where the goal-conditioned policy $\pi^G(\mathbf{a} \mid \mathbf{s}, \mathbf{g})$ navigates toward a goal $\mathbf{g}$, and (2) the "**Explore-phase**," where the exploration policy $\pi^E(\mathbf{a} \mid \mathbf{s})$ gathers new state trajectories to refine the goal-conditioned policy.

While Go-Explore relies on task-specific information for goal selection and executes random actions for exploration, our method uses task-agnostic temporal distance metrics induced by temporal distance-aware representations. The subsequent sections detail how our method leverages the temporal distance-aware representations for selecting goals in the Go-phase (Section 3.4), enhancing the exploration policy (Section 3.5), and facilitating the GCRL policy training (Section 3.6).

### 3.4 Exploratory Goal Selection

For unsupervised GCRL, selecting low-density (less visited) states as exploratory goals can enhance goal-directed exploration [15, 16]. However, the concept of the "density" of a state does not necessarily indicate how rare or hard it is to reach the state. For example, while a robotic arm might actively seek out unseen (low-density) joint positions, interacting with objects could offer more significant learning opportunities [29]. Thus, we propose selecting goals that are *temporally distant* from states that are already visited (i.e. in the replay buffer) to explore not only diverse but also hard-to-reach states.

To sample a faraway goal at the start of each episode, we employ the non-parametric particle-based entropy estimator [27] on top of our temporal distance-aware representations. Among states in a

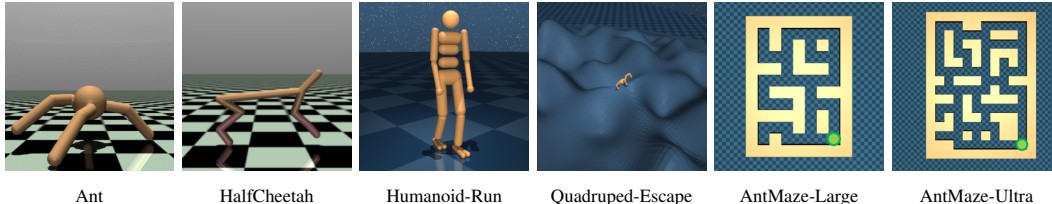

Figure 3: We evaluate our method on 6 state-based robotic locomotion environments.

minibatch, we choose $N$ goals with high entropy and collect $N$ corresponding trajectories using the goal-reaching policy. The entropy can be estimated as follows, which we refer to as *TLDR reward*:

$$r_{\text{TLDR}}(\mathbf{s}) = \log \left( 1 + \frac{1}{k} \sum_{\mathbf{z}^{(j)} \in N_k(\phi(\mathbf{s}))} \|\phi(\mathbf{s}) - \mathbf{z}^{(j)}\| \right), \tag{2}$$

where $N_k(\cdot)$ denotes the $k$-nearest neighbors around $\phi(\mathbf{s})$ within a minibatch.

### 3.5 Learning Exploration Policy

After the goal-conditioned policy navigates towards the chosen goal $\mathbf{g}$ for $T_G$ steps, the exploration policy $\pi_\theta^E$ is executed to discover states even more distant from the visited states. This objective of the exploration policy can be simply defined as:

$$r^E(\mathbf{s}, \mathbf{s}') = r_{\text{TLDR}}(\mathbf{s}') - r_{\text{TLDR}}(\mathbf{s}). \tag{3}$$

Similar to LEXA [10], we alternate between goal-reaching episodes and exploration episodes. For goal-reaching episodes, we execute the goal-conditioned policy until the end of the episodes. For exploration episodes, we sample the timestep $T_G \sim \text{Unif}(0, T - 1)$ at the beginning of each episode and execute the exploration policy if the current timestep $t \geq T_G$.

### 3.6 Learning Goal-Conditioned Policy

The goal-conditioned policy aims to minimize the distance to the goal. However, defining "distance" to the goal often requires domain knowledge. Instead, we propose leveraging a task-agnostic metric, temporal distance, as the learning signal for the goal-conditioned policy:

$$r^G(\mathbf{s}, \mathbf{s}', \mathbf{g}) = \|\phi(\mathbf{s}) - \phi(\mathbf{g})\| - \|\phi(\mathbf{s}') - \phi(\mathbf{g})\|. \tag{4}$$

If our representations accurately capture temporal distances between states, optimizing this reward in a greedy manner becomes sufficient for learning an optimal goal-reaching policy.

## 4 Experiments

In this paper, we propose TLDR, a novel unsupervised GCRL method that utilizes temporal distance-aware representations for both exploration and optimizing a goal-conditioned policy. Through our experiments, we aim to answer the following three questions: (1) Does TLDR explore better compared to other exploration methods? (2) Is our goal-conditioned policy better than prior unsupervised GCRL methods? (3) How crucial is TLDR for goal-conditioned policy learning and exploration?

### 4.1 Experimental Setup

**Tasks.** As illustrated in Figure 3, we evaluate TLDR in 6 state-based environments: **Ant** and **HalfCheetah** from OpenAI Gym [30], **Humanoid-Run** and **Quadruped-Escape** from DeepMind Control Suite (DMC) [31], **AntMaze-Large** from D4RL [32], and **AntMaze-Ultra** [33]. For Humanoid-Run and Quadruped-Escape, we include the 3D coordinates of the agents in their observations. In addition, we also evaluate on two pixel-based environments: **Quadruped (Pixel)** from METRA [1] and **Kitchen (Pixel)** from D4RL [32], with the $64 \times 64 \times 3$ image observation.

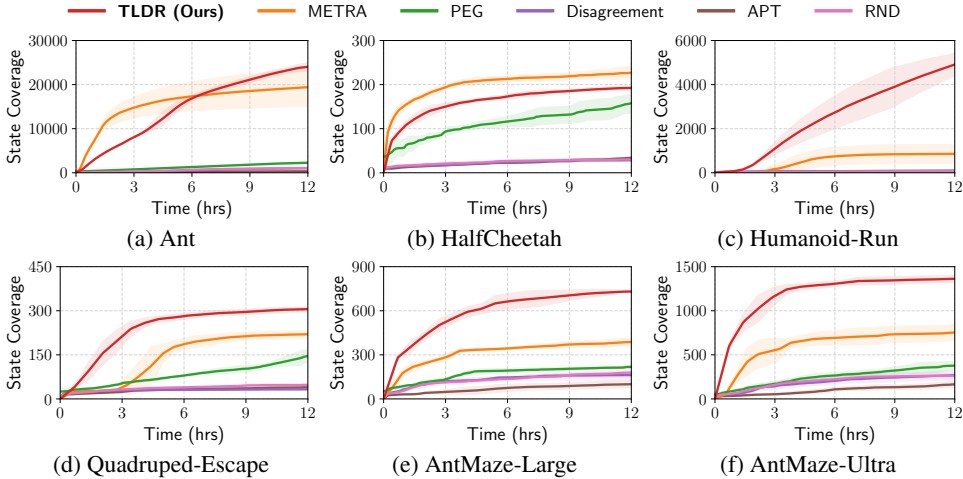

Figure 4: **State coverage in state-based environments.** We measure the state coverage of unsupervised exploration methods. Our method consistently shows superior state coverage compared to other methods, except in HalfCheetah compared against METRA.

**Comparisons.** We compare our method with 6 prior unsupervised GCRL, skill discovery, and exploration methods. For state-based environments, we compare with METRA, PEG, APT, RND, and Disagreement. For pixel-based environments, we compare with METRA and LEXA.

- **METRA** [1]: leverages temporal distance-aware representations for skill discovery.
- **PEG** [2]: plans to obtain goals with maximum exploration rewards.
- **LEXA** [10]: uses world model to train an Achiever and Explorer policy.
- **APT** [27]: maximizes the entropy reward estimated from the $k$-nearest neighbors in a minibatch.
- **RND** [34]: uses the distillation loss of a network to a random target network as rewards.
- **Disagreement** [35]: utilizes the disagreement among an ensemble of world models as rewards.

## 4.2 Quantitative Results

In Figure 4, we compare the state coverage during training (i.e. the number of $1 \times 1$ sized $(x, y)$-bins occupied by any of the training trajectories). TLDR outperforms all prior works, except in HalfCheetah compared to METRA. METRA learns low-dimensional skills and focuses on extending the temporal distance along a few directions specified by the skills, providing a strong inductive bias for simple locomotion tasks like HalfCheetah. On the other hand, TLDR achieves much larger state coverage in complex environments than METRA, including AntMaze-Large, AntMaze-Ultra, and

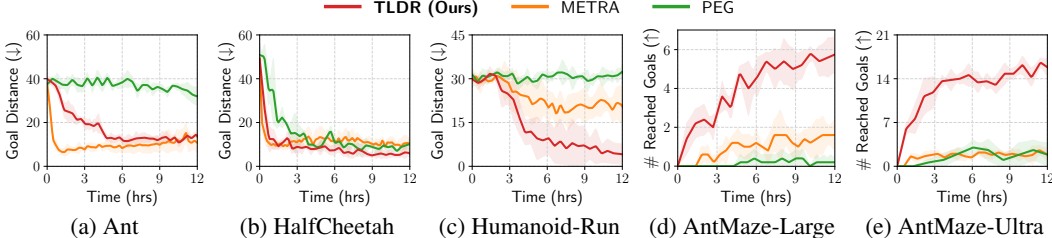

Figure 5: **Goal-reaching metrics of a goal-conditioned policy.** For (a) Ant, (b) HalfCheetah, and (c) Humanoid-Run, we report the average distance between goals and the last states of trajectories (lower is better). TLDR achieves a comparable average goal distance to METRA. For AntMaze environments, we report the number of pre-defined goals reached by a goal-reaching policy (7 for (d) AntMaze-Large and 21 for (e) AntMaze-Ultra), and TLDR significantly outperforms prior works.

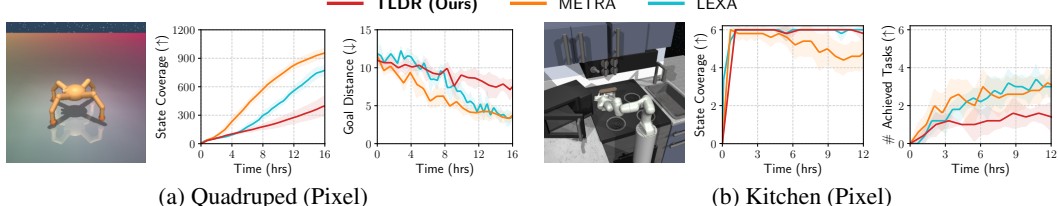

(a) Quadruped (Pixel)  (b) Kitchen (Pixel)

Figure 6: **Results in pixel-based environments.** We compare TLDR with prior works in pixel-based Quadruped and Kitchen environments. In Quadruped (Pixel), TLDR demonstrates a slow learning speed compared to METRA and LEXA. For Kitchen (Pixel), TLDR could interact with all six objects during training but shows low success rates for evaluation.

Quadruped-Escape, where all other methods struggle and only explore limited regions. This shows the strength of our method in the exploration of complex environments.

We then compare the goal-reaching performance of TLDR with PEG and METRA in Figure 5 by measuring the average distance between goals and the last states of trajectories. The results in Figures 5a to 5c show that TLDR can navigate towards the given goals closer than or at least on par with METRA. Figures 5d and 5e show that TLDR is the only method that can navigate towards a various set of goals in both mazes, demonstrating its superior exploration and goal-conditioned policy learning with temporal distance.

In Appendix B, we further show the comparisons in environment steps, not in hours. PEG shows better sample efficiency in relatively low-dimensional or easy-exploration tasks, such as Ant and HalfCheetah. However, the state coverages of PEG quickly converge to narrower regions, especially in AntMazes, than those of TLDR. METRA generally shows worse sample efficiency than TLDR.

Figure 6 shows the results in pixel-based environments. In Quadruped (Pixel), TLDR explores diverse regions but learns slower than LEXA and METRA. For Kitchen (Pixel), TLDR interacts with all six objects during training, but struggles at learning the goal-conditioned policy. Further analysis in Appendix F suggests that the performance bottleneck is related to goal-conditioned policy learning with pixel observations. We leave more detailed analyses for future works.

### 4.3 Qualitative Results

Figure 7 visualizes the learned goal-reaching behaviors on the AntMaze-Ultra environment. TLDR can successfully reach both near and far-away goals in diverse regions. On the other hand, METRA and PEG fail to navigate to diverse goals. METRA could reach some goals distant from the initial position, whereas PEG fails to reach temporally faraway goals. This clearly shows the benefit of using temporal distance in unsupervised GCRL. More qualitative results can be found in Appendix D.

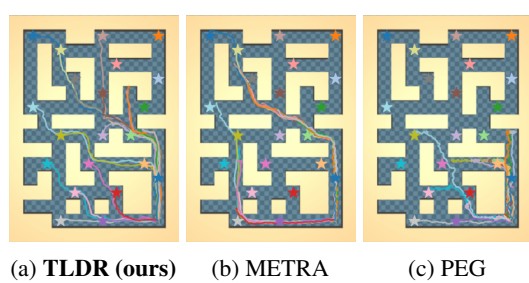

(a) **TLDR (ours)**  (b) METRA  (c) PEG

Figure 7: TLDR can cover more goals compared to METRA and PEG in AntMaze-Ultra.

### 4.4 Ablation Studies

To investigate the importance of temporal distance-aware representations in our algorithm, we conduct ablation studies on GCRL reward designs and exploration strategies.

**GCRL reward design.** We compare with three different goal-conditioned policy learning methods: (1) QRL [14], which learns a quasimetric value function and latent dynamics model, (2) sparse HER [8], which uses the sparse goal-reaching reward $-\mathbb{1}(\mathbf{s} \neq \mathbf{g})$, and (3) DDL [19], which uses

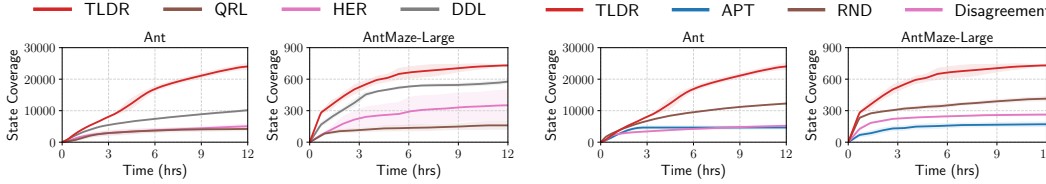

(a) TLDR with different GCRL rewards        (b) TLDR with different exploration methods

Figure 8: We evaluate our method with different design choices for (a) GCRL rewards and (b) exploration methods on Ant and AntMaze-Large. TLDR shows better state coverages than its ablated versions in both ablation studies, indicating the importance of using temporal distance-aware representations for both exploration and GCRL.

expected temporal distances as rewards. Figure 8a and Figure 21 show the superior performance of our temporal distance-based GCRL reward over HER and DDL, suggesting the importance of using optimal temporal distance as a dense reward signal. Furthermore, although QRL learns a value function that preserves optimal temporal distances, it struggles to learn an effective goal-reaching policy. Unlike QRL, which directly uses the learned value function along with an additional latent dynamics model, TLDR leverages temporal distance-aware representations to compute dense rewards for the goal-conditioned policy and shows better performance. In Appendix G, we show that this trend also holds with a fixed dataset, which ignores the effect of exploration and only compares goal-reaching reward designs for goal-reaching performances.

**Exploration strategy.** For goal selection and exploration rewards, we replace TLDR reward in Equation (2), with other exploration bonuses: APT (with ICM [36] representations), RND, and Disagreement. Note that goal-conditioned policies are still trained with the same temporal distance-based rewards as TLDR, thereby comparing only exploration strategies. As shown in Figure 8b, using TLDR reward for goal selection and exploration rewards achieves significantly higher performance than other exploration bonuses. This result implies that our temporal distance-based rewards are effective for unsupervised exploration.

## 5   Conclusion

In this paper, we introduce TLDR, an unsupervised GCRL algorithm that incorporates temporal distance-aware representations. TLDR leverages temporal distance for exploration and learning the goal-reaching policy. By pursuing states with larger temporal distances, TLDR can continuously explore challenging regions, achieving better state coverage. The experimental results demonstrate that TLDR can cover significantly larger state spaces across diverse environments than existing unsupervised RL algorithms.

**Limitations.** While TLDR achieves remarkable state coverages, it still has several limitations:

- TLDR shows a slow learning speed compared to METRA in *pixel-based environments*. Our analysis in Appendix F demonstrates the need for further research in representation learning and GCRL for pixel observations.
- Our temporal distance-aware representations do not capture *asymmetric temporal distance* between states, which can make policy learning challenging for highly asymmetric environments.
- Applying unsupervised RL to *real robots* has many challenges, including safety. While not tested on real robots, our preliminary results in Appendix E indicate that combining TLDR with safety-aware techniques [37, 38] is a promising future direction for real robotic systems.
- TLDR achieves high efficiency in terms of wall clock time, but not in terms of *sample efficiency*, as shown in Appendix B. We believe that increasing the update-to-data ratio or using model-based RL could enhance the sample efficiency of our method.

**Acknowledgments**

This work was supported in part by the Institute of Information & Communications Technology Planning & Evaluation (IITP) grant (RS-2020-II201361, Artificial Intelligence Graduate School Program (Yonsei University)), the National Research Foundation of Korea (NRF) grant (RS-2024-00333634), and the Electronics and Telecommunications Research Institute (ETRI) grant (24ZR1100) funded by the Korean Government (MSIT).

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

# A Training Details

## A.1 Computing Resources and Experiments

All experiments are done on a single RTX 4090 GPU and 4 CPU cores. Each state-based experiment takes 12 hours for all methods, following METRA [1], which trains each method for 10-12 hours. We report the number of environment steps used for the methods in our experiments in Table 1. We use 5 random seeds for all experiments and report the mean and standard deviation of the results.

Table 1: The number of environment steps for experiments.

| Environment | TLDR | METRA | PEG | LEXA | APT | RND | Disagreement |
|---|---|---|---|---|---|---|---|
| Ant | 56.5M | 83.2M | 0.7M | - | 2.4M | 4.1M | 4.8M |
| HalfCheetah | 51.4M | 103.5M | 0.7M | - | 2.5M | 4.2M | 5.0M |
| AntMaze-Large | 42.6M | 62.5M | 0.7M | - | 2.4M | 6.4M | 5.0M |
| AntMaze-Ultra | 31.2M | 44.5M | 0.6M | - | 2.4M | 4.5M | 3.4M |
| Quadruped-Escape | 28.0M | 34.8M | 0.6M | - | 2.2M | 4.5M | 4.4M |
| Humanoid-Run | 40.8M | 59.9M | 0.6M | - | 3.5M | 4.7M | 4.7M |
| Quadruped (Pixel) | 3.9M | 4.1M | - | 2.1M | - | - | - |
| Kitchen (Pixel) | 1.1M | 1.7M | - | 1.0M | - | - | - |

## A.2 Implementation Details

Our method, TLDR, is implemented on top of the official implementation of METRA. Similar to METRA, we use SAC [39] for learning the goal-reaching policy and exploration policy. We train our temporal distance-aware representation $\phi(\mathbf{s})$ by maximizing the following objective:

$$\mathbb{E}_{\mathbf{s} \sim p_\mathbf{s}, \mathbf{g} \sim p_\mathbf{g}} \left[ f(\|\phi(\mathbf{s}) - \phi(\mathbf{g})\|) + \lambda \cdot \min\left(\epsilon, 1 - \|\phi(\mathbf{s}) - \phi(\mathbf{s}')\|\right) \right], \quad (5)$$

where $f$ is an affine-transformed softplus function:

$$f(x) = -\text{softplus}(500 - x, \beta = 0.01), \quad (6)$$

which prevents the distances $\|\phi(\mathbf{s}) - \phi(\mathbf{g})\|$ from diverging, following QRL [14].

For training the exploration policy, we normalize the TLDR reward used in Equation (3) to keep the rewards on a consistent scale. We simply divide the TLDR reward by a running estimate of its mean value, following APT [27].

For METRA, PEG, and LEXA, we use their official implementation. For random exploration approaches (APT, RND, Disagreement), we use the implementation from URLB [40].

## A.3 Hyperparameters

The hyperparameters used in our experiments are summarized in Table 2.

For METRA, we use 2-D continuous skills for Ant, 16-D discrete skills for HalfCheetah, 24-D discrete skills for Kitchen (Pixel), and 4-D continuous skills for other environments. We use the batch size of 1024 for state-based environments and 256 for pixel-based environments. We set the number of gradient steps per epoch for each experiment to be the same as ours. We use the default values for the remaining hyperparameters. To perform goal-reaching tasks with METRA, we set the skill $\mathbf{z}$ as $\frac{\phi(\mathbf{g}) - \phi(\mathbf{s})}{\|\phi(\mathbf{g}) - \phi(\mathbf{s})\|}$ for continuous skills or $\arg\max_{\text{dim}} (\phi(\mathbf{g}) - \phi(\mathbf{s}))$ for discrete skills.

In PEG, we use the same hyperparameters used in their AntMaze experiments. Since PEG uses the normalized goal space, we measure the range of the observations and normalize the goal states according to the minimum and maximum range.

In LEXA, we follow their hyperparameters and opt for the temporal distance reward for training the Achiever policy.

For APT (with ICM encoder), RND, and Disagreement, we use the same hyperparameters as in URLB [40].

For the ablation with QRL, we use the learning rate of 0.0003 for the critic. We use an (input dim)-1024-1024-128 network for the encoder, 256-1024-2048 for the projector, IQE-maxmean head of 64 components of size 32, and 128-1024-1024-128 for the latent dynamics model. The transition loss is weighted by 1. For HER, we use the discount factor $\gamma = 0.99$.

Table 2: List of hyperparameters.

| Hyperparameter | Value |
|---|---|
| Learning rate | 0.0001 |
| Learning rate for $\phi$ | 0.0005 |
| Batch size | 1024 (State), 256 (Pixel) |
| Replay buffer size | $10^6$ (State), $3 \times 10^5$ (Quadruped (Pixel)), $10^5$ (Kitchen) |
| Frame stack (Pixel) | 3 |
| Optimizer | Adam [41] |
| Relaxation constant $\epsilon$ in Eq. (5) | $10^{-3}$ |
| $\dim \phi(\mathbf{s})$ | 8 (Kitchen), 4 (Others) |
| $k$ in Eq. (2) | 12 |
| Initial $\lambda$ | $3 \times 10^3$ |
| SAC entropy coefficient | 0.01 (Kitchen), target entropy as $(-\dim \mathcal{A})/2$ (others) |
| Discount factor $\gamma$ | 0.97 (Goal-reaching policy), 0.99 (Exploration policy) |
| Normalization | LayerNorm [42] for the critics, None for $\phi$ and actors |
| Encoder for image observations | CNN |
| MLP dimensions | 1024 |
| MLP depths | 2 |
| Goal relabelling | 0.8 (sampled from future observations), 0.2 (no relabelling) |
| # of gradient steps per epoch | 50 (Ant, HalfCheetah, Humanoid-Run, Quadruped-Escape), 75 (AntMaze-Large), 100 (Kitchen (Pixel)), 150 (AntMaze-Ultra), 200 (Quadruped (Pixel)) |
| # of episode rollouts per epoch | 8 |
| $\tau$ for updating the target network | 0.995 |

## A.4  Environment Details

**Ant.**  We use the MuJoCo Ant environment in OpenAI gym [30]. The observation space is 29-D and the action space is 8-D. Following METRA, we normalize the observations for Ant with a fixed mean and standard deviation of observations computed from randomly generated trajectories. The episode length is 200.

**HalfCheetah**  We use the MuJoCo HalfCheetah environment in OpenAI gym [30]. The observation space is 18-D and the action space is 6-D. Following METRA, we normalize the observations for HalfCheetah with a fixed mean and standard deviation of observations from randomly generated trajectories. The episode length is 200.

**Humanoid-Run.**  We use the Humanoid-Run task from DeepMind Control Suite [31]. The global $x, y, z$ coordinates of the agent are added to the observation. Humanoid has 55-D observation space with 21-D action space. The episode length is 200.

**Quadruped-Escape.**  Quadruped-Escape is included in DeepMind Control Suite [31]. The quadruped robot is initialized in a basin surrounded by complex terrains. Due to the complex terrains, moving further away from the initial position is challenging. Similar to the AntMaze environments, we fix the terrain shape. Also, we add the global $x, y, z$ coordinates of the agent to the observation. Quadruped-Escape has 104-D observation space with 12-D action space. The episode length is 200.

**AntMaze-Large.**  We use `antmaze-large-play-v2` in D4RL [32]. The observation and action spaces are the same as the Ant environment. The episode length is 300. To make exploration more challenging, we fix the initial location of the agent to be the bottom right corner of the maze, as shown in Figure 3 (AntMaze-Large).

**AntMaze-Ultra.**  We use `antmaze-ultra-play-v0` proposed by Jiang et al. [33]. The observation and action spaces are the same as the Ant environment. The episode length is 600. Similar to AntMaze-Large, we fix the initial location of the agent to be the bottom right corner of the maze, as shown in Figure 3 (AntMaze-Ultra).

**Quadruped (Pixel).**  We use the pixel-based version of the Quadruped environment [31] used in METRA [1]. Specifically, we use the image size of $64 \times 64 \times 3$ with episode length of 200.

**Kitchen (Pixel).**  We use the pixel-based version of the Kitchen environment [43] used in ME-TRA [1] and LEXA [10]. Specifically, we use the image size of $64 \times 64 \times 3$ with the episode length of 50. The action space has 9 dimensions.

### A.5 Evaluation Protocol

For Ant, Humanoid, and Quadruped (Pixel), we sample goals with $(x, y)$-coordinates from $[-50, 50]^2$, $[-40, 40]^2$, and $[-15, 15]^2$, respectively. For the rest of the goal state (e.g. joint poses), we use the initial robot configuration following Park et al. [1].

For HalfCheetah, we sample goals with $x$-coordinates from $[-100, 100]$.

For AntMaze-Large and AntMaze-Ultra, we use the pre-defined goals as shown in Figure 7. A goal is deemed to be reached when an ant gets closer than 0.5 to the goal.

For Kitchen (Pixel), we use the same 6 single-task goal images used in LEXA [10], which consist of interactions with Kettle, Microwave, Light switch, Hinge cabinet, Slide cabinet, and Bottom burner. We report the total number of achieved tasks during evaluation.

For all environments, we use a full state as a goal. Specifically, for state-based observations, we use the observation upon reset as the base observation and switch the $x, y$ coordinates (or $x$ for HalfCheetah) to the corresponding dimensions. For Quadruped (Pixel), we render the image of the state where the agent is at the goal position and use it as the goal.

For each environment, state coverage is calculated by the number of $1 \times 1$-sized $(x, y)$-bins $((x)$-bins for HalfCheetah) occupied by any of the training trajectories. For Kitchen (Pixel), the state coverage is calculated as the number of tasks achieved at least once during the last 100000 environment steps.

## B  Sample Efficiency Comparison

We compare the sample efficiency of TLDR, METRA and PEG under the same setting as in Section 4.2. Since PEG requires longer training time for the same amount of environment steps compared to TLDR, PEG is trained for more than 72 hours, while TLDR and METRA are each trained for 12 hours.

Figures 9 and 10 illustrate the state coverage and goal-reaching metrics with respect to the number of environment steps used for training. TLDR exhibits superior state coverages and goal-reaching performance in hard-exploration tasks like AntMazes. In contrast, PEG tends to be more sample-efficient in environments with relatively low-dimensional state and action spaces, such as Ant and HalfCheetah, but it quickly converges to narrower regions in other environments that require hard-exploration (AntMazes) or have higher-dimensional state and action spaces (Quadruped-Escape, Humanoid-Run). METRA shows worse sample efficiency overall compared to TLDR.

PEG's exploration via latent disagreement may over-prioritize less critical dimensions of the state spaces (e.g., joint angles) and may not scale well to high-dimensional observation spaces. Additionally,

its reliance on expected temporal distances can be less effective for training a goal-reaching policy than TLDR's optimal temporal distances, as shown in Section 4.4. Moreover, METRA's skill learning objective can incentivize revisiting known distant states rather than exploring new ones, leading to suboptimal convergence.

Efficient exploration in high-dimensional spaces remains a major challenge in learning complex real-world tasks. Unlike other methods that quickly converge to suboptimal solutions in these settings, TLDR effectively handles this challenge and continues to improve steadily. We believe that increasing the update-to-data ratio or incorporating model-based reinforcement learning approaches could further enhance TLDR's sample efficiency.

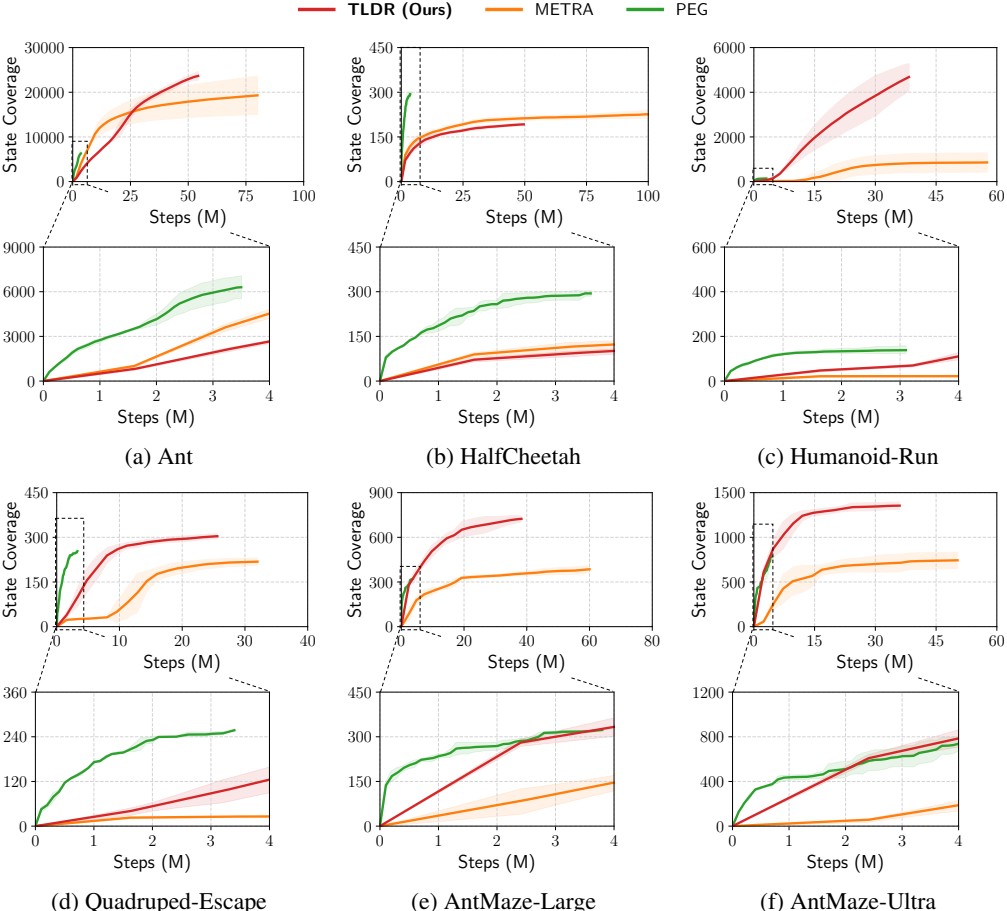

Figure 9: **State coverage in state-based environments (sample efficiency).** We plot the state coverage in terms of the environment steps used for training. PEG is trained for >72 hours for comparison. PEG, as a model-based GCRL algorithm, is more sample efficient for relatively low-dimensional tasks like Ant or HalfCheetah but struggles to learn in more challenging environments such as AntMaze. METRA is generally less sample efficient compared to TLDR.

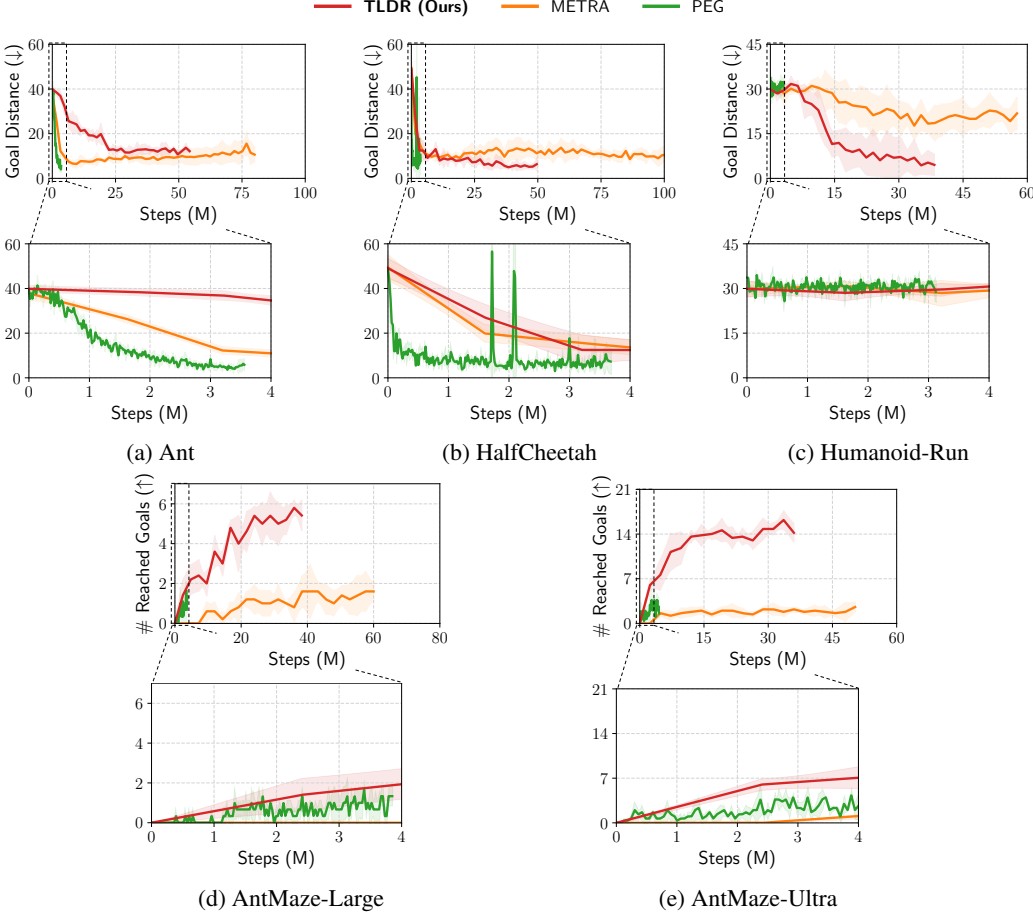

Figure 10: **Goal-reaching metrics of a goal-conditioned policy (sample efficiency).** Similar to the results in Figure 9, while PEG can be trained efficiently in relatively low-dimensional tasks, TLDR has better sample complexity in more challenging tasks.

# C  More Ablation Studies

We conduct the ablation studies on the number of nearest neighbors $k$ (Figure 11) and $\dim \phi(\mathbf{s})$ (Figure 12) used in Equation (2). Figure 11 shows that in Ant environment, $k = 12$ provides the best results, with exploration slightly degrading at $k = 5$ or 20; in the AntMaze-Large environment, the performance is rarely affected by the changes in $k$. Regarding $\dim \phi(\mathbf{s})$, the performance is nearly the same across different settings. Our main experimental results in Section 4.2 use $k = 12$ and $\dim \phi(\mathbf{s}) = 4$, which demonstrates robust performance across diverse environments.

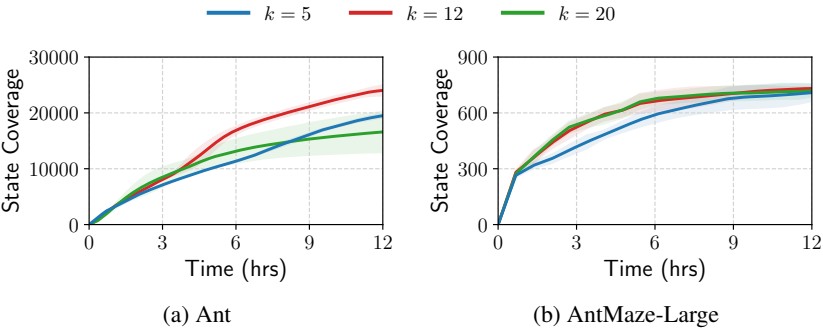

|  (a) Ant  |  (b) AntMaze-Large |

Figure 11: **State coverage on state-based environments with different** $k$**.** We measure the state coverage of our method with $k \in \{5, 12, 20\}$ used for calculating the TLDR reward in Equation (2). For Ant, $k = 12$ works the best. For AntMaze-Large, $k$ does not affect the final state coverage.

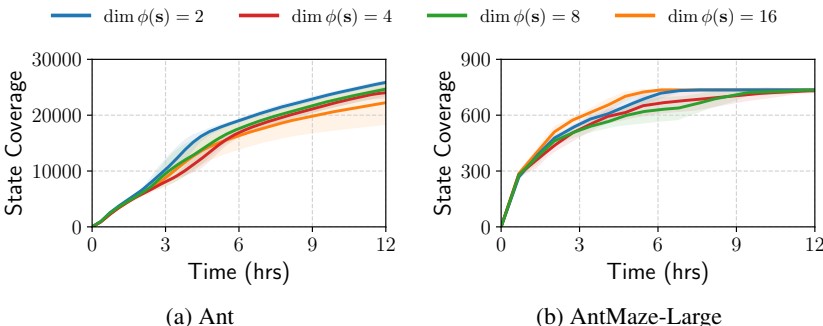

|  (a) Ant  |  (b) AntMaze-Large |

Figure 12: **State coverage on state-based environments with different** $\dim \phi(\mathbf{s})$**.** We measure the state coverage of our method with $\dim \phi(\mathbf{s}) \in \{2, 4, 8, 16\}$, where $\dim \phi(\mathbf{s})$ is the dimension of the temporal distance-aware representations. The results show that $\dim \phi(\mathbf{s})$ does not have a critical impact on the performance in these environments.

## D   More Qualitative Results

We include more qualitative results in Figures 13 to 16. For the qualitative results in Quadruped-Escape (Figure 14), we evenly select 48 states satisfying $x^2 + y^2 = 10^2$, where $x, y$ represents the agent position. The $z$ coordinate is selected as the minimum possible height that the agent does not collide with the terrain. For all environments, TLDR achieves the best goal-reaching behaviors compared to the other unsupervised GCRL methods, covering the goals in more diverse regions.

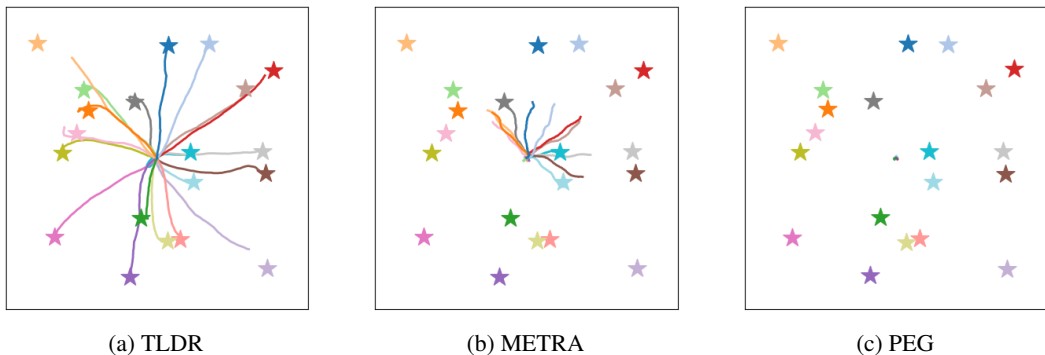

(a) TLDR                    (b) METRA                    (c) PEG

Figure 13: **Goal-reaching ability in Humanoid-Run.** We evaluate each method with the goals sampled according to Appendix A.5. TLDR moves further towards the goal in diverse directions compared to other methods.

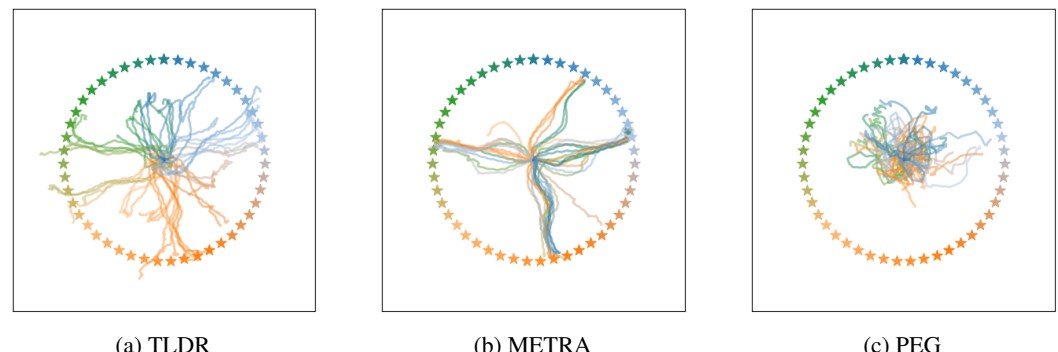

(a) TLDR                    (b) METRA                    (c) PEG

Figure 14: **Goal-reaching ability in Quadruped-Escape.** We evaluate each method with the goals that are evenly selected at the same distance from the origin. TLDR can not only cover more regions but also have a better goal-reaching capability than other methods.

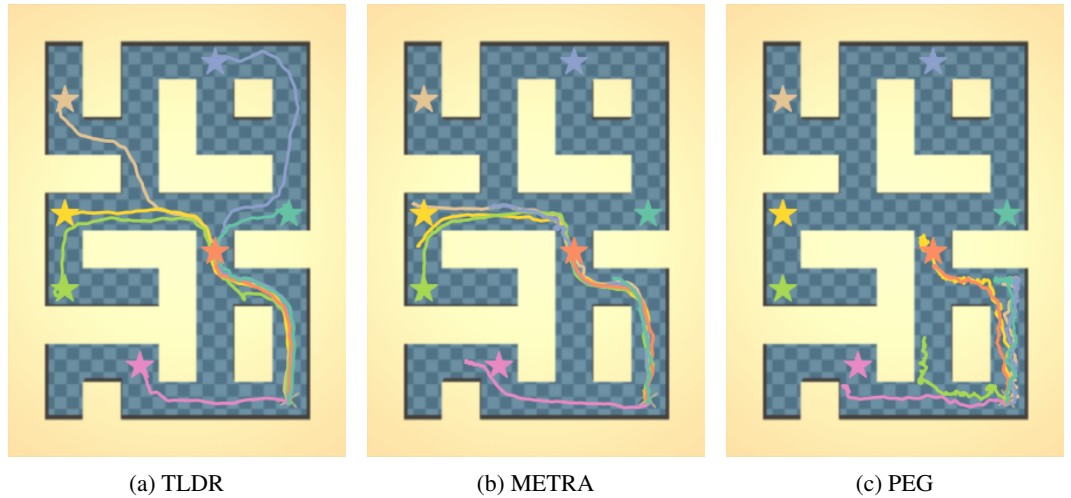

|            |             |           |
|:----------:|:-----------:|:---------:|
| (a) TLDR   | (b) METRA   | (c) PEG   |

Figure 15: **Goal-reaching ability in AntMaze-Large.** TLDR can reach most of the goals in AntMaze-Large, while other GCRL methods struggle to reach distant goals.

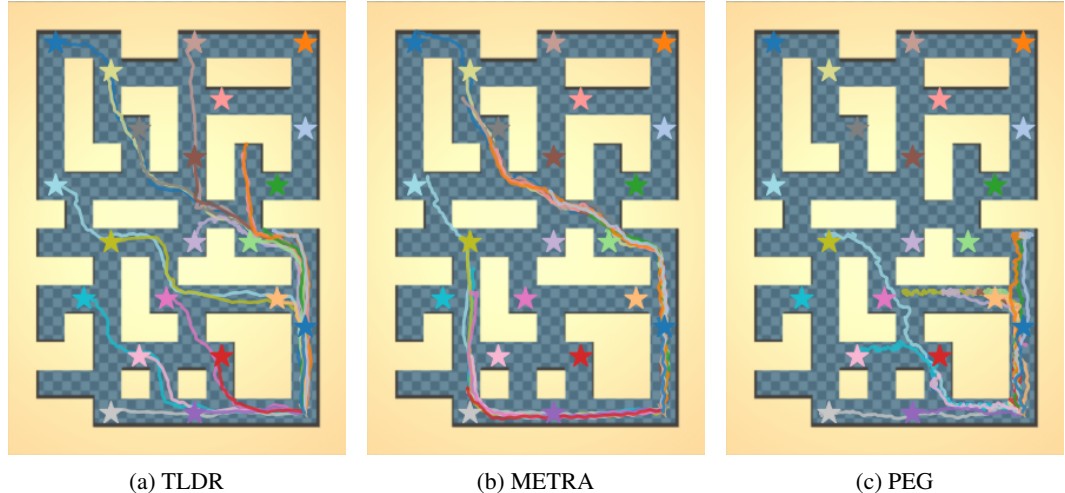

|            |             |           |
|:----------:|:-----------:|:---------:|
| (a) TLDR   | (b) METRA   | (c) PEG   |

Figure 16: **Goal-reaching ability in AntMaze-Ultra.** Similar to Figure 15, TLDR can cover the most number of goals in AntMaze-Ultra, outperforming other methods.

# E Unitree A1 Simulation Results

While most unsupervised goal-conditioned RL and skill discovery research currently focuses on simulated environments, unsupervised RL holds great potential for learning emergent and efficient skills for real-world robots.

To investigate whether TLDR can explore in environments with real-world robotic counterparts, we train TLDR on the Unitree A1 robot in simulation [44] (Figure 17), considering sim-to-real transfer approaches. TLDR and METRA are trained for 6 hours, with the same hyperparameter settings we used in Quadruped-Escape and the episode length of 200. For goal-conditioned evaluation, we sample goals with $(x, y)$-coordinates from $[-15, 15]^2$.

As shown in Figure 18a and Figure 18b, TLDR achieves substantially better state coverage and goal-reaching performance compared to METRA, suggesting the potential of TLDR for autonomous exploration and effective goal-reaching gaits learning in real robotics systems.

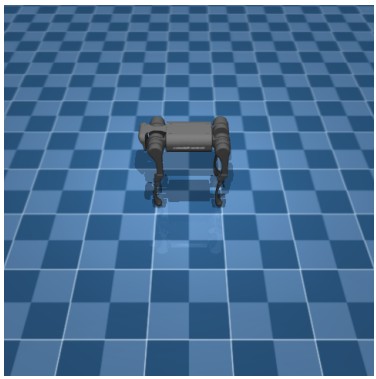

Figure 17: **Unitree A1 Simulation Environment.** To demonstrate the potential applicability for real-world robots, we choose the environment that simulates a Unitree A1 robot with 12 DoFs.

However, the learned behaviors with TLDR might be unsafe to transfer to reality since it does not impose any constraint on the learned behaviors beyond the goal-reaching objective. To address this, we test incorporating a safety reward for learning the exploration and goal-conditioned policies. The safety reward is defined as $r_{safe} = [0, 0, 1] \cdot \mathbf{v}_{torso}$, where $\mathbf{v}_{torso}$ is the orientation of robot torso, which equals to $[0, 0, 1]$ when the robot is upright.

The results in Figure 18 show that TLDR with this safety reward can match the performance of TLDR without regularization in terms of state coverage and goal-reaching metrics while also maximizing the safety reward. These findings indicate that TLDR is compatible with additional reward signals, and applying advanced safety-aware techinques [37, 38] could facilitate the learning of safer behaviors. Videos on learned behaviors can be found at `https://heatz123.github.io/tldr`.

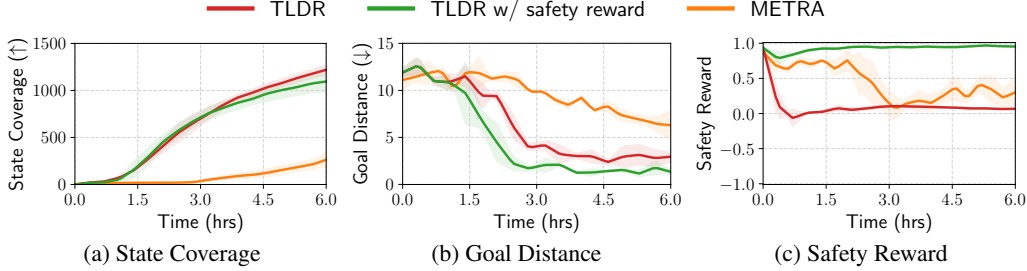

(a) State Coverage    (b) Goal Distance    (c) Safety Reward

Figure 18: **Learning curves of Unitree A1 Simulation Results.** TLDR achieves better state coverage (a) and goal-reaching performance (b) compared to METRA. Since the learned behavior can be unsafe, we consider the setting that the safety reward (c) is given by $r_{safe} = [0, 0, 1] \cdot \mathbf{v_{torso}}$, where $\mathbf{v_{torso}}$ is the orientation of robot torso which equals to $[0, 0, 1]$ with upright direction. Even with adding the reward, TLDR can (a) still explore the state space and (b) learn effective goal-reaching behaviors (c) while maximizing the safety reward.

# F   Analysis on Pixel-based Environments

While achieving remarkable exploration and goal-reaching performance in state-based environments, exploration slows down in pixel-based environments, as observed in Figure 6. To identify the performance bottleneck for learning in pixel-based settings, we compare the performance when replacing pixel observations with state observations for the inputs of goal-conditioned policy, exploration policy, and TLDR encoder, respectively.

Figure 19 shows that the performance of TLDR becomes comparable to METRA when state observations are used instead of pixel observations for the goal-reaching policy, while other modifications do not improve upon the original TLDR. This suggests that the main bottleneck for exploration is likely to be the representations for the goal-conditioned policy. Based on this result, a promising future direction for improving TLDR in pixel-based environments could be the integration of advanced representation learning techniques [45, 46, 47] into our learning pipeline.

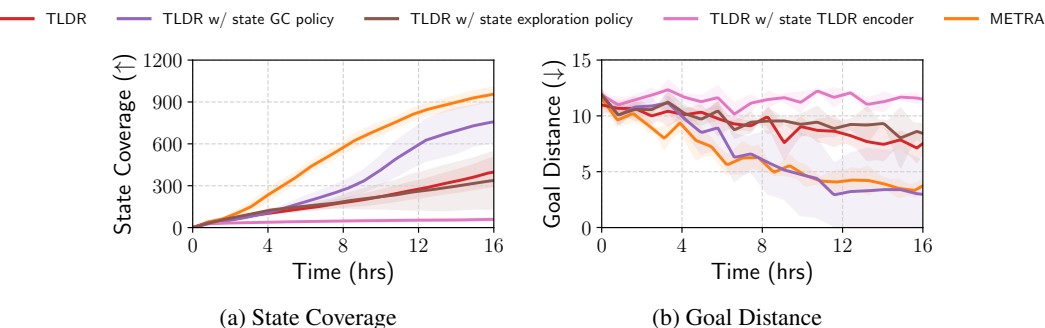

(a) State Coverage    (b) Goal Distance

Figure 19: **Result of component-wise analysis in Quadruped (Pixel).** To identify the bottleneck of exploration with pixel observations, we swap pixel observations to state observations for the input of goal-conditioned policy, exploration policy, and TLDR encoder, respectively. Exploration of TLDR significantly improves when we input state observations to the goal-conditioned policy, which suggests the main bottleneck for exploration is the representations for the goal-conditioned policy.

Additionally, while METRA quickly learns to achieve skills in Kitchen (Pixel) environment compared to TLDR, we observe that its performance degrades with continuous skills, as shown in Figure 20. This implies the difficulty of learning policies with continuous goals in TLDR's goal-conditioned policy learning, possibly because learning to reach arbitrary states is more challenging than mastering a specific set of discrete behaviors. Investigating ways to restrict the size of the goal space in TLDR could be an interesting direction for future research.

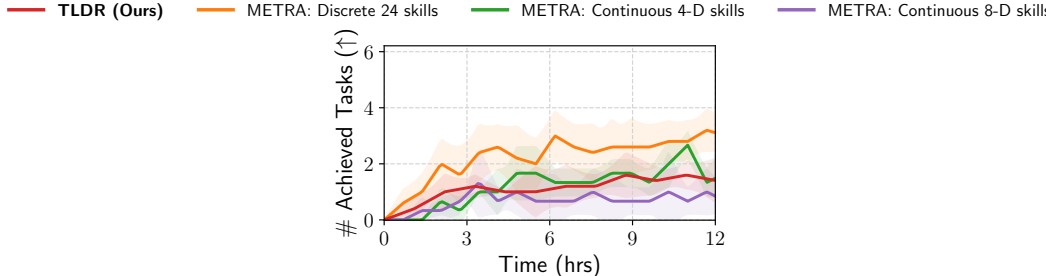

Figure 20: **Performance of METRA in Kitchen (Pixel) with different skill settings.** When METRA uses continuous skill vectors in Kitchen (Pixel), METRA's performance substantially degrades with continuous skills in Kitchen (Pixel) environment, which is similar to our setting of learning to reach any goals.

# G Analysis on Goal-reaching Reward Design

We compare the goal-reaching performance of TLDR with different goal-conditioned policy learning methods using the same experimental setup as in Figure 8a. As presented in Figure 21, other goal-reaching policy learning methods cannot reach the same level of performance as TLDR. This highlights the importance of our GCRL reward design on goal-reaching performance, consistent with the results in Figure 8a, where TLDR achieves the best state coverage while other methods struggle.

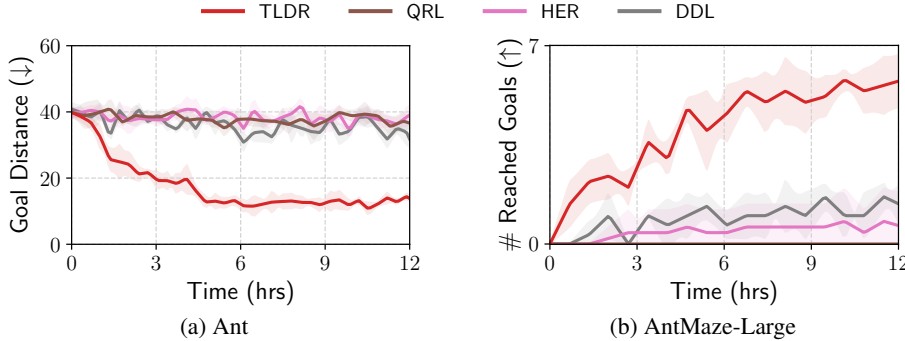

(a) Ant                                      (b) AntMaze-Large

Figure 21: **Goal-reaching metrics with GCRL reward design ablations.** TLDR shows better goal-reaching performance compared to other choices of goal-conditioned policy learning methods, showing the effectiveness of our design of the GCRL reward.

To further isolate the impact of the exploration strategy and focus solely on goal-reaching policy learning, we evaluate the goal-reaching performance in an offline learning setting. In this setup, policies are trained on a fixed dataset of 1M samples collected from rollouts of a trained TLDR policy.

Although goal-reaching performances are degraded in this setting due to off-policy training, our choice of the goal-reaching reward still demonstrates superior results compared to other methods, as shown in Figure 22. This suggests that our design of the goal-reaching reward—minimizing the L2 distance to the goal in the temporal distance-aware representation space—provides effective signals for goal-reaching.

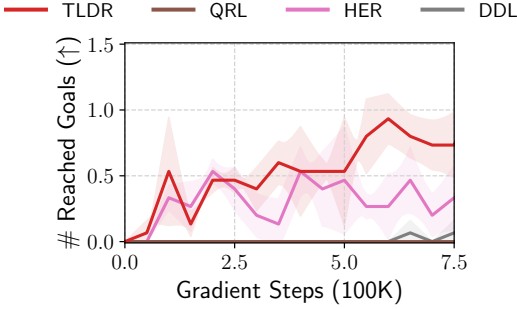

Figure 22: **Goal-conditioned policy learning ablation with fixed dataset.** With 1 million samples of rollouts from a trained TLDR policy, we learn the goal-conditioned policy without adding the data to the replay buffer, differing only in the goal-conditioned policy learning methods. TLDR shows the best performance in this setting where the impact of exploration is isolated.

