# OpenReview forum: "TLDR: Unsupervised Goal-Conditioned RL via Temporal Distance-Aware Representations"
_robot-learning.org/CoRL/2024/Conference — CoRL 2024_

### Official Review · Reviewer_nYH6 · 2024-07-16
**TLDR: a method for unsupervised goal-conditioned exploration**

**Originality:** 3
**Technical Quality:** 3
**Clarity Of Presentation:** 4
**Potential Impact:** 3
**Recommendation:** 3
**Confidence:** 3

**Review:**

Strengths
- The problem the paper is trying to tackle, scalable exploration in unsupervised goal-conditioned reinforcement learning, is very relevant.
- The method achieves excellent state coverage in many complex state-based environments, outperforming prior unsupervised GCRL and skill discovery methods
- The goal-reaching policy significantly outperforms prior work on AntMaze-environments
- The method is evaluated on various simulated environments and compared to recent baselines.
- The method is very lightweight and efficient to run, reaching a competitive number of environment steps for experiments in limited wall-clock time.

Weaknesses
- The work is arguably incremental, given that the idea of using temporal distances for learning representations was present in the METRA paper [1]. I would summarize the paper's main idea as eliminating the skill vectors in METRA and replacing them with explicit goals (please correct me if my understanding is incorrect). However, this requires training an explicit goal-reaching policy, which METRA does not have. Furthermore, having to formalize a goal in the state space can be a restrictive assumption in real-world robotics, especially in the case of image-based inputs or locomotion environments. However, this is a fairly common issue with GCRL methods, so I will not decrease the paper grade because of this.
- In terms of goal success rate and distance in state-based environments, TLDR only improves the results significantly in AntMaze (Fig. 10), which raises some questions. For instance, TLDR achieves better state coverage than the baselines in Humanoid-Run. However, what does that ultimately tell us about how TLDR will perform in downstream tasks? Is state coverage the metric we should use to measure the exploration quality?
- The performance of the method in pixel-based environments (Fig. 7) is underwhelming, both in terms of state coverage and goal success rate
- Lack of theoretical algorithm analysis or convincing theoretical insights into why this method should outperform prior work.
- Lack of real-world robotic experiments. I see at least two hypothetical problems. First, the lack of scalability in image-based environments is a big issue. Second, selecting exploration goals as far away as possible might lead to oscillatory, unstable, or unsafe behavior.
- Other evaluation metrics than wall-clock time would be desirable for real-world robotics. Sample efficiency is an obvious choice, but also something like the number of hard resets (human interventions).

[1] S. Park, O. Rybkin, and S. Levine. Metra: Scalable unsupervised RL with metric-aware abstraction. In International Conference on Learning Representations, 2024

**Quality Of The Limitations Section:**

3

**Questions For Rebuttal:**

- Eq 4: Are the signs of the terms incorrect? For example, assuming that s' == g, the first component of the reward will be zero, and the reward will be equal to the negative distance between s and g. Note that the code (L1646 in metra.py) has the signs the other way around.
- More insight into the downstream (goal-reaching) performance would benefit the paper. Ultimately, that is why we care about exploration. For instance, why does the superior state coverage in Humanoid-Run not convert into better downstream performance (Fig. 5)?
- Fig 5b vs Fig 10b discrepancy. In Fig 5b, TLDR outperforms METRA and PEG on HalfCheetah, whereas in Fig 10b, the methods are approximately equal. Which is correct, or are the plots measuring something different?
- Did you analyze or have any intuitions as to why the performance in pixel-based environments is lacking?
- Will you open-source the code if the paper is accepted?

**Robotics Focus:**

2

**Summary Of Paper:**

Prior unsupervised goal-conditioned reinforcement learning (GCRL) methods struggle to achieve good exploration in complex environments. The authors propose TLDR, an unsupervised GCRL method that uses temporal distance-aware representations to learn exploration and goal-reaching policies. The exploration policy of TLDR tries to explore states far away from states that have already been visited, whereas the goal-reaching policy learns to minimize the distance to the goal. The experiments in simulated robotic environments demonstrate that TLDR achieves excellent state coverage in the benchmark environments.

**Summary Of Recommendation:**

Getting pure exploration methods to reliably scale to complex environments with high intrinsic dimensionality has been a challenge, and TLDR suffers from the same issue. Given that the method in its current form lacks clear theoretical inspiration, struggles in image-based environments, is not evaluated on real-world robotics, and the goal-reaching performance in state-based environments is good but not great, I cannot recommend accepting this paper in its current form, despite the excellent state coverage achieved in state-based environments. Edit: updated to weak accept after author response.

---

### Official Review · Reviewer_7NjC · 2024-07-20
**A valid work, but not suited for a robotic conference**

**Originality:** 3
**Technical Quality:** 4
**Clarity Of Presentation:** 4
**Potential Impact:** 3
**Recommendation:** 2
**Confidence:** 4

**Review:**

I think this is a valid work which, though, would be more suited for a conference that mainly focuses on reinforcement learning or machine learning/artificial intelligence.
## Strengths
* **Method**: the idea presented is sensible and original. The results of applying the approach are interesting and useful, despite the decrease in performance when working with pixels-based environments
* **Presentation**: the presentation is overall clear and enjoyable. There are some minor remarks concerning the presentation of some results (see Questions for Rebuttal)
## Weaknesses
* **Unsuitability**: I don't think this work is well suited for CORL. The work presents some improvements in terms of exploration and goal-reaching performance in simulation, with no real-world validation (lack of Outcome, see [here](https://www.corl.org/contributions/instruction-for-reviews)). There is no major focus on problems that are currently relevant to robotics or robot learning (lack of Intent, see [here](https://www.corl.org/contributions/instruction-for-reviews)). See the Questions section for additional feedback on this.
* **Performance on visual environments**: the authors clearly show that the method fails to perform as well as other approaches on pixel-based environments. While this could be addressed in future work, providing an intuition or some additional experiments to show why that's the case would be useful to understand better the representation learned by TLDR

**Quality Of The Limitations Section:**

2

**Questions For Rebuttal:**

* Why do the results in the Figures presented use "time" as the x-axis? It's more common (and a more sensible choice) to adopt the number of steps in the environment
* To get better insights into the performance of using TLDR as a representation for goal reaching, it would be useful to show results like the ones presented in Figure 5, but to be obtained by training on the same dataset for all the goal-reaching approaches (removing the dependence of the exploration strategy)
* It would be useful to have some more insights or ideas on why the approach is failing in pixel-based settings
* In order to show intent, I believe the work should have addressed problems or techniques relevant for robotics. While GCRL could potentially be applied for robotics, works that have a robotic focus tend to show how to apply such systems in real world in a stable way, e.g. [1,2]. Another way the authors could have shown intent, it's by benchmarking on real robotic systems in simulation, e.g. using the assets from [3]. The environments shown (apart, maybe, for the Franka one) have no real counterparts and are used to test out general approaches for control, not because you would expect the algorithms to transfer to real-world systems. There are other suites that are more appropriate for that purpose, e.g. [4,5].
* Exploration in robotics is generally considered unsafe and the work does not address this issue (it doesn't even discuss it as a limitation). Given that a significant part of the contribution comes from better exploration performance, the problem of (un)safe exploration should be addressed or at least discussed

[1] Stabilizing Contrastive RL: Techniques for Robotic Goal Reaching from Offline Data, Zheng et al

[2] From Play to Policy: Conditional Behavior Generation from Uncurated Robot Data, Cui et al

[3] https://github.com/google-deepmind/mujoco_menagerie

[4] https://github.com/google-research/realworldrl_suite

[5] HumanoidBench: Simulated Humanoid Benchmark for Whole-Body Locomotion and Manipulation, Sferrazza et al

**Robotics Focus:**

2

**Summary Of Paper:**

The work presents TLDR, which learns temporal distance aware representations to improve exploration and goal-conditioned behaviors.

**Summary Of Recommendation:**

It's a valid work, with a novel method and interesting results, but it's clearly not a good fit for a robotic conference, given the absence of focus on robotics problems and/or real-world experiments

---

### Official Review · Reviewer_uLZM · 2024-07-21
**Submission339 Review**

**Originality:** 2
**Technical Quality:** 3
**Clarity Of Presentation:** 3
**Potential Impact:** 3
**Recommendation:** 3
**Confidence:** 3

**Review:**

The TLDR approach significantly improves exploration by selecting goals that are temporally distant from the current state. This ensures that the agent covers a larger state space, which is crucial in complex environments where previous methods often fall short. By utilizing temporal distance as a reward signal, TLDR provides a more informative and dense learning signal compared to traditional sparse or binary rewards. This helps in training more effective goal-conditioned policies. The method is unsupervised and does not rely on task-specific knowledge or external supervision. This makes it a scalable approach for pre-training robots and other intelligent agents, as it can be applied to various environments without requiring manual intervention.

The calculation of temporal distances and the subsequent learning of distance-aware representations can be computationally intensive. This might limit the method's scalability to very large state spaces or real-time applications. Authors do mention this.

Reference to some prior work is missing - Dynamical Distance Learning for Semi-Supervised and Unsupervised Skill Discovery (Kristian Hartikainen) et al. Discussion about how is this different thatn LEXA[10] would be helpful.

**Quality Of The Limitations Section:**

3

**Questions For Rebuttal:**

Reference to some prior work is missing - Dynamical Distance Learning for Semi-Supervised and Unsupervised Skill Discovery (Kristian Hartikainen) et al. Discussion about how is this different thatn LEXA[10] would be helpful
Is there a reason you did not compare with these methods?
The explanation of the ablation experiments is can be expanded. Especially, the reasoning behind plot 8b.

**Robotics Focus:**

2

**Summary Of Paper:**

The paper introduces a novel approach to unsupervised goal-conditioned reinforcement learning (GCRL). The proposed method addresses the challenges of limited exploration and sparse or noisy rewards by leveraging TemporaL Distance-aware Representations (TLDR). TLDR enhances both exploration and goal-reaching capabilities by selecting distant goals and using temporal distances as rewards. This strategy allows the agent to explore a wider range of states and achieve better goal-reaching performance. Experimental results in various simulated robotic environments demonstrate that TLDR significantly outperforms some previous methods in state coverage and efficiency.

**Summary Of Recommendation:**

Would like the authors to address some of the comments questions in order to be more confident about acceptance.

---

### Author Rebuttal · Authors · 2024-08-10

We thank all reviewers for their constructive feedback! The attached file includes our revised paper and a video. Below we briefly summarize the main points in our rebuttal:

&nbsp;

**Applicability to robotics and safety concerns.**

In Appendix, Section D of the revised paper, we include the training TLDR on the Unitree A1 robot in simulation. Figure 18 (b),(c) shows that TLDR is able to explore and discover goal-reaching gaits without hand-engineered rewards. We also conducted experiments incorporating safety considerations by penalizing unstable poses, as shown in Figure 18, demonstrating the potential of TLDR for safe exploration.

We believe the improvements made with TLDR are crucial stepping stones towards real-world implementation and merit discussion at CoRL.

&nbsp;

**Performance analysis on visual environments.**

We have analyzed the performance bottleneck of TLDR in pixel-based environments in Appendix, Section E of the revised paper. Our additional experiments identify that TLDR’s challenges stem from training the goal-conditioned policy $\pi(a \vert s, g)$ with high-dimensional pixel observations and continuous goal representations, suggesting that the performance of TLDR with pixel observations could improve with advanced representation learning methods for goal-conditioned policy learning.

&nbsp;

**Comparison with DDL, LEXA and METRA.**

We have elaborated the comparison with DDL, LEXA and METRA in the related work section.  We also included comparisons to goal-conditioned reward designs of DDL in Figure 10(b), 21, 22. The results show that our design of the temporal distance was crucial for better goal-conditioned policy learning.

&nbsp;

**Comparison with sample efficiency.**

We have included plots on both “steps” and “time” in the updated Figure 4, 6 (time) and 5, 7 (steps). TLDR generally shows better sample efficiency compared to METRA, and PEG’s state coverages quickly converge to narrower regions in challenging environments like AntMaze.

&nbsp;

**Ablations on the choice of our GC reward design.**

We included an ablation study for TLDR on the design of goal-conditioned rewards with prior goal-reaching approaches using the same dataset. Figure 22 of the revised paper shows that our design of goal-conditioned reward shows the highest performance among the comparisons.

&nbsp;

We hope we have addressed all your concerns and questions. Please let us know if there are any concerns preventing you from raising your score.

---

### Decision · Program_Chairs · 2024-09-04

**Decision:**

Accept

**Comment:**

This paper introduces novel ideas that are shown to be effective on some tasks. There are several concerns: Most importantly for CoRL, its applicability to robotics is unclear. While the methods are potentially relevant to robotics, it is evaluated entirely in simulation. The most robotics-related task is the Kitchen environment but this plays a less-central role in the paper, and TLDR fails to show superior performance. Sim2Real transfer is not discussed. Thus, arguably this work is out of scope for CoRL. Secondly, TLDR performs poorly in pixel-based environments; this would merit further discussion. Thirdly, the methods should be more explicitly contrasted with LEXA and METRA.

During the rebuttal, the authors clarified how TLDR relates to LEXA and METRA. They provided additional experimental results on visual environments. Without solving the issue of lower performance, they do shed some light on the issue.

This paper falls into the *Gray* area of CoRL. It clearly does not address the *Outcome* criterion, and the *Intent* arguably very weakly. However, it does hold promise for eventual application in robotics; thus, in my judgment, it should not be rejected on the grounds of being out of scope.